# CONFIDENTIAL-PROFITT:
# CONFIDENTIAL PROof OF FaIr TRAINING OF TREES

**Ali Shahin Shamsabadi[1], Sierra Wyllie[2,3], Nicholas Franzese[4], Natalie Dullerud[2,3]**
**Sébastien Gambs[\*5], Nicolas Papernot[\*2,3], Xiao Wang[\*4], Adrian Weller[\*1,6]**
[1] The Alan Turing Institute, [2] University of Toronto, [3] Vector Institute, [4] Northwestern University,
[5] Université du Québec à Montréal, [6] University of Cambridge

## ABSTRACT

Post hoc auditing of model fairness suffers from potential drawbacks: (1) auditing may be highly sensitive to the test samples chosen; (2) the model and/or its training data may need to be shared with an auditor thereby breaking confidentiality. We address these issues by instead providing a certificate that demonstrates that the learning algorithm itself is fair, and hence, as a consequence, so too is the trained model. We introduce a method to provide a confidential proof of fairness for training, in the context of widely used decision trees, which we term Confidential-PROFITT. We propose novel fair decision tree learning algorithms along with customized zero-knowledge proof protocols to obtain a proof of fairness that can be audited by a third party. Using zero-knowledge proofs enables us to guarantee confidentiality of both the model and its training data. We show empirically that bounding the information gain of each node with respect to the sensitive attributes reduces the unfairness of the final tree. In extensive experiments on the COMPAS, Communities and Crime, Default Credit, and Adult datasets, we demonstrate that a company can use Confidential-PROFITT to certify the fairness of their decision tree to an auditor in less than 2 minutes, thus indicating the applicability of our approach. This is true for both the demographic parity and equalized odds definitions of fairness. Finally, we extend Confidential-PROFITT to apply to ensembles of trees.

## 1 INTRODUCTION

The deployment of machine learning models in high-stake decision systems (Waddell, 2016; Benjamens et al., 2020; Kleinberg et al., 2018) is associated with the risk of unfair decisions towards particular subgroups defined by sensitive attributes (Dwork et al., 2012). A canonical approach for auditing such deployment is to measure the fairness of a trained model on a reference dataset (Pentyala et al., 2022). In practice, this would be done by an external party (i.e., an *auditor*). Such audits, however, can be difficult to organize and are sensitive to the choice of reference dataset (Fukuchi et al., 2020). This may lead to a form of unhelpful interaction between the company and the auditor, in which the company could deny a model is unfair by claiming that the reference dataset does not belong to the training distribution used, or the auditor can forge a reference dataset that could be used to blame the company for unfair predictions. One avenue to address this problem would be for the company to release its training data and the model to the auditor who can then verify that a fair training algorithm was used by e.g., locally rerunning the training process. However, this approach does not protect the confidentiality of the company's training data.

In this paper, we remediate these issues by introducing confidential proofs of fair training. We highlight that our method does not *guarantee* fairness. Rather, our approach employs a tunable parameter controlling the resulting degree of fairness. The certificate we provide proves that our approach was employed, and also includes the specific parameter value used, and the resulting fairness metrics on the training data. We call this approach "fairness-aware training", or "fair training" for short. Concretely, we design a framework (i.e., Confidential-PROFITT) that allows a company to directly prove to the auditor, through the execution of a cryptographic protocol, that the *learning algorithm* used to train the model was fair by design. To achieve this without revealing the company's

---

*Contributed Equally

dataset or model to the auditor, we rely on zero knowledge (ZK) proofs (Goldwasser et al., 1985; Goldreich et al., 1991), which allow a party to prove statements about their private data without revealing it. Our framework is generic in the sense that it allows the company to prove to interested parties (e.g., users in addition to the auditor) the fairness of the model learning process—thus increasing public trust in the model.

We instantiate the confidential proof of fair training in the context of decision trees, which are widely used by companies in sensitive domains such as healthcare (Podgorelec et al., 2002) and finance (Ghatasheh, 2014; Güntay et al., 2022), in part due to their performance, in their ability to be leveraged in ensemble methods (e.g.,, random forests), and also, occasionally, due to their assumed interpretability (Molnar, 2020). We propose a cryptographic protocol that can prove the fair training of decision trees with various common fairness definitions including demographic parity (Calders et al., 2009) and equalized odds (Hardt et al., 2016). Fair decision tree learning builds a decision tree iteratively by splitting its dataset according to a criterion capturing the information gain with respect to both the class and sensitive attributes. In particular, our criterion addresses issues of prior fair training algorithms by connecting information gain with respect to the sensitive attribute to demographic parity.

To render the verification more efficient, once it is implemented cryptographically, we propose a co-design involving concepts originating from both machine learning and cryptography. First, we design a ZK-friendly fair decision tree learning algorithm based on the insight that verifying fairness is sufficient because the accuracy is of no interest for the auditor. Prior works (Kamiran et al., 2010; Raff et al., 2018) on fair decision tree learning consider only demographic parity and view the problem as as a joint optimization problem between accuracy and demographic parity. The optimal solution is obtained by searching exhaustively through an enumeration of all possible attributes and split points. However, in ZK, the whole exhaustive search would need to be proven using heavy cryptographic machinery, which is computationally expensive in most settings of interest. To address this, we propose a generalized framework that formulates training as a constrained optimization problem, in which fairness is represented as a constraint and the accuracy is the objective of optimization. As a result, we only need to verify the satisfaction of the constraint in ZK without verifying the accuracy. Second, we propose an efficient ZK protocol that can perform the aforementioned fairness constraint verification efficiently using state-of-the-art ZK protocols in the RAM model (Franzese et al., 2021). Instead of using a naïve verification approach that requires proving a computation as long as the training process, we design a fairness verification protocol with complexity sublinear to the training phase.

In summary, we propose a novel way of auditing model fairness by proving directly that the training algorithm itself is fair rather than inspecting the model and its predictions. As a consequence, we do not need a reference dataset and both the training data and the model remain confidential, i.e., are not disclosed to the auditor. We highlight the following contributions:

1. We propose a new ZK-friendly fair decision tree learning algorithm such that its fairness can be verified efficiently without repeating the entire training process. We summarize existing fairness metrics and show that our approach is generally applicable to these metrics.
2. We design and implement a specialized ZK proof protocol to efficiently verify the fairness of the above training algorithm. While prior works exist in *secure inference* using ZK proofs, to the best of our knowledge we are the first to propose *secure training* using ZK proofs.
3. We implement and evaluate our framework in terms of accuracy, fairness, running time, and scalability using decision trees and random forests trained on a variety of real-world datasets. For example on the Community and Crime dataset, Confidential-PROFITT can provide a certificate in less than 9 seconds that the company employed our fair training algorithm and obtained a decision tree. In this example, our fair training algorithm can improve the equalized odds fairness, which is not supported by prior works (Kamiran et al., 2010; Raff et al., 2018), by 50% with negligible effects on the accuracy. In our implementation, Confidential-PROFITT provides the same certificate for random forests in 70 seconds.

## 2    PROBLEM STATEMENT, BACKGROUND, AND RELATED WORK

**Problem statement.** Ignoring fairness in the training process may result in models that negatively affect users belonging to specific subgroups with respect to a sensitive attribute such as gender, race or disability (Kamiran & Calders, 2009; Raff et al., 2018). For example, Buolamwini & Gebru (2018) demonstrated that darker-skinned women are the most misclassified demographic group in

commercially-used facial recognition systems. Similarly, Amazon's automated recruiting system displayed a negative bias against hiring women[1]. Therefore, it is important for companies to demonstrate that their models have been trained with fairness constraints associated with minimizing negative biases. Our objective in this work is to provide a certificate proving that a company employs a fair training algorithm with a tunable parameter controlling the resulting degree of fairness[2]. This is challenging for several reasons. First, the sharing of data, collected predominantly from users, with the third party might not be permitted due to the associated privacy risks (Shokri & Shmatikov, 2015). Second, confidentiality issues also arise as companies are usually not willing to disclose their models to protect their intellectual property (Zhang et al., 2018). Thus, the proof of fairness should ensure *both* the confidentiality of the training data as well as that of the model.

More formally, we consider two parties: a *prover* (i.e., company, with a private training dataset $D = \{X, Y\}$ in which $X$ and $Y$ denote respectively, the set of dataset attributes and ground-truth labels) and a *verifier* (i.e., auditor). Both the prover and verifier must agree on the dimensions of $(X, Y)$ and the sensitive attribute $a$, which is one of the columns of $X$, and a fair decision tree training algorithm $A(\cdot, \cdot)$ (in practice the prover is likely to be the entity deciding these parameters as they are also responsible for training the model). For the sake of simplicity, we assume that $Y$ and $a$ are binary and we use DT $\leftarrow A(X, Y)$ to denote the resulting decision tree. Our Confidential PROof of FaIr Tree Training (Confidential-PROFITT) enables the prover to prove to the verifier that it has a DT model, which is indeed correctly computed as $A(X, Y)$. See Appendix A for a table of our notations.

**Fairness.** Many fairness metrics have been defined based on different philosophical and moral assumptions (Heidari et al., 2019; Narayanan, 2018). To show the generalizability of our framework, we consider four common notions of group fairness. *Demographic Parity* ignores the ground-truth label and ensures the equal probability of a predicted positive label across subgroups. In comparison, *Equalized Odds* ensures that the predicted label is conditionally independent of the sensitive attribute and the ground-truth label. *Equality Opportunity* and *Predictive Equality* enforce a partial form of equalized odds, suitable in domains in which being included (or not) in the positive class is viewed respectively as a *desirable* or *poor* outcome. Hereafter, we give the definitions for the first two metrics, with more details available in Appendix B:

**Definition 1** (Demographic Parity (Calders et al., 2009)). *A predictor $\hat{Y}$ satisfies Demographic Parity with respect to the sensitive attribute $a$ if:*

$$\Pr[\hat{Y} = 1 | a = 0] = \Pr[\hat{Y} = 1 | a = 1] \qquad \forall 0, 1 \in a.$$

**Definition 2** (Equalized Odds (Hardt et al., 2016)). *A predictor $\hat{Y}$ satisfies Equalized Odds with respect to the sensitive attribute $a$ if:*

$$\Pr[\hat{Y} = 1 | Y = y, a = 0] = \Pr[\hat{Y} = 1 | Y = y, a = 1] \qquad \forall y \in \{0, 1\}, \forall 0, 1 \in a.$$

**Fairness and decision trees.** Fair decision tree algorithms are usually either based on greedy optimizers (Kamiran et al., 2010; Loh, 2011) or Mixed-Integer Programming (MIP) solvers (Bennett, 1992; Aghaei et al., 2019; Jo et al., 2022). The former optimizes a fairness-accuracy splitting criterion for each node based on the data routed to it while the latter solves a mathematical optimization augmented with a fairness constraint. In this paper, we focus on greedy-based optimization methods as MIP-based methods are currently only applicable on datasets with at most thousands of inputs and with non-continuous features (Zantedeschi et al., 2021). One such greedy approach from Kamiran et al. (2010) defines a fair training algorithm by reformulating the information gain traditionally used for decision tree training to encourage demographic parity fairness. This work was extended to random forests by Raff et al. (2018). Both works manage the accuracy-fairness trade-off via a novel gain term encouraging discrimination over the class label while discouraging discrimination with respect to the sensitive attribute, leading to both high performance and fairness.

**Zero-knowledge proof.** We design a Zero-Knowledge (ZK) proof protocol that lets the prover prove to the verifier that their model is trained using the fair training algorithm without revealing

---

[1] https://www.reuters.com/article/us-amazon-com-jobs-automation-insight/amazon-scraps-secret-ai-recruiting-tool-that-showed-bias-against-women-idUSKCN1MK08G

[2] See Practical considerations in Section 6 for a discussion of how the company can subsequently prove that the fair-trained mode is being deployed.

their data or model, thus preserving confidentiality. Given an agreed upon program $P$, a ZK proof protocol $\Pi$ enables a prover to convince a verifier that they possess an input $w$ such that $P(w) = 1$, while revealing no additional information about $w$ (Goldwasser et al., 1985; Goldreich et al., 1991). Typically, a ZK proof protocol $\Pi$ has the following properties:

- *Completeness* – For any input $w$ that $P$ evaluates to 1 in the clear, an honest prover (who behaves correctly) can convince an honest verifier that $P(w) = 1$ using $\Pi$.
- *Soundness* – Given an input $w$ that $P$ does *not* evaluate to 1 in the clear, no malicious prover (who can behave arbitrarily) can falsely convince an honest verifier that $P(w) = 1$ using $\Pi$.
- *Zero Knowledge* – If the prover and verifier execute $\Pi$ to prove that $P(w) = 1$, even a malicious verifier (who can behave arbitrarily) learns no information about $w$ other than what can be inferred from the fact that $P(w) = 1$.

In the context of this paper, we write a fairness verification program that takes as inputs a decision tree and a dataset and evaluates to 1 if and only if the decision tree is trained using the fair training algorithm $A(\cdot, \cdot)$ on the dataset. We instantiate this ZK proof protocol from a vector-oblivious linear evaluation (Weng et al., 2021), with a recent extension to support ZK RAM accession (Franzese et al., 2021). Circuit-based models of computation incur relatively high overhead when the control flow of a program depends on the input (Goldreich & Ostrovsky, 1996). This limitation often becomes a bottleneck when writing programs for decision trees (e.g., when selecting a particular attribute of a sample to test against the threshold in each node). Utilizing a RAM model ZK bypasses much of this overhead, thereby allowing for a succinct and efficient implementation of our fairness verification program.

**Cryptographic protocols and decision trees.** Prior works (Hamada et al., 2021; Abspoel et al., 2021; Adams et al., 2021) designed protocols to allow multiple parties, each with a private dataset, to jointly train a decision tree. These protocols work in a different setting compared to ours, where all private inputs come from the company proving the correctness of private computation. Other works (Zhang et al., 2020a) developed ZK proofs for decision trees at inference but not at training time.

## 3 CONFIDENTIAL-PROFITT

We propose Confidential-PROFITT to confidentially certify the fair training of a decision tree. Confidential-PROFITT consists of four main steps (see also the block diagram in Appendix C):

0. **Local training:** The prover *locally* trains a decision tree on their private training data using our proposed fair decision tree learning algorithm. The algorithm is designed to be both i) *crypto-friendly* and efficient in ZK via a constraint optimization problem that maximizes the accuracy gain while the constraining the unfairness gain to some threshold; and ii) *generic* in supporting many fairness metrics (described in Section 2) by defining information gain with respect to each fairness metric.
1. **Data and model commitment:** The prover commits to the training data and the decision tree trained on this data. These commitments are binding and hiding, meaning that the prover cannot change the content being committed without the verifier's consent and that the commitment does not reveal anything about the underlying content (Katz & Lindell, 2007).
2. **Proving counter update in ZK:** The prover locally computes and commits to the path each data point takes through the trained decision tree. The prover proves in ZK that the committed paths are correct by comparing the values of each data point to the splitting values in the tree. Then for each value of the sensitive attribute $a$, the prover counts the number of data points with value $a$ that pass through each node, and proves that these counts are valid.
3. **Prove fairness constraint in ZK:** The prover computes fairness-related information gain of each node using their corresponding committed counters and proves to the verifier that this is below the desired threshold.

Next, we describe our fair decision tree learning algorithm and our ZK protocol in detail.

### 3.1 FAIR DECISION TREE LEARNING ALGORITHM

In this section, we propose a novel training algorithm to learn fair decision trees. The decision tree is learned by recursively partitioning the training dataset (Breiman et al., 2017) (see Appendix D for a detailed description of tree training). Splitting by the value, val, of the attribute, attr, partitions the

---

**Algorithm 1:** Finding the best split for each node using our fair learning algorithm.

---

**Input:** Dataset $X$ in $j$-th node DT [j], Sensitive attribute $a$, Threshold $\tau$ over unfairness-based information gain $Gini_{\text{Unfairness}}(\cdot)$.
**Output:** Best split

---

1:  Threshold_attribute_gains = []          ▷ Initialize gains for each value of each attribute
2:  Parent = $X$                                                              ▷ Make parent node
3:  **for** attr $\in$ Attributes **do**
4:     **for** $x \in X$ **do**
5:        val = $x$[attr]                                         ▷ Set split threshold value per datapoint
6:        Child$_1$ = $\{x \in X | x[\text{attr}] < \text{val}\}$, Child$_2$ = $\{x \in X | x[\text{attr}] \geq \text{val}\}$            ▷ Make children
7:        $Gain_{\text{Accuracy}} = Gini_{\text{Accuracy}}(\text{Parent}) - \sum_{k=1}^{2} \frac{|\text{Child}_k|}{|\text{Parent}|} Gini_{\text{Accuracy}}(\text{Child}_k)$                    ▷ Gain wrt label
8:        $Gain_{\text{Unfairness}} = Gini_{\text{Unfairness}}(\text{Parent}) - \sum_{k=1}^{2} \frac{|\text{Child}_k|}{|\text{Parent}|} Gini_{\text{Unfairness}}(\text{Child}_k)$     ▷ Gain wrt sensitive attribute
9:        **if** $Gain_{\text{Unfairness}} \leq \tau$ **then**
10:           Threshold_attribute_gains.append(attr, val, $\{Gain_{\text{Accuracy}}, Gain_{\text{Unfairness}}\}$)                ▷ Record fair splits
11:  **return** DT[$j$].attr, DT[$j$].val, DT[$j$].gains = $\max_{Gain_{\text{Accuracy}}}$ Threshold_attribute_gains          ▷ Best split

---

dataset at the parent node into two *children* nodes, Parent = $\{\text{Child}_1, \text{Child}_2\}$, such that:

$$\text{Child}_1 = \{x \in \text{Parent} | x[\text{attr}] < \text{val}\}, \quad \text{Child}_2 = \{x \in \text{Parent} | x[\text{attr}] \geq \text{val}\}. \tag{1}$$

In general, the information gain of each split, $Gain(\text{Split})$, is the amount of information improved in the children nodes with respect to their *parent* node:

$$Gain(\text{Split}) = Gini(\text{Parent}) - \sum_{k=1}^{2} \frac{|\text{Child}_k|}{|\text{Parent}|} Gini(\text{Child}_k), \tag{2}$$

in which $Gini$ quantifies the impurity of each node. We introduce two different variants of $Gini$: for accuracy-based information ($Gini_{\text{Accuracy}}$) and for unfairness-based information ($Gini_{\text{Unfairness}}$) to capture the effect of each split on both accuracy and fairness.

We define $Gini_{\text{Accuracy}}$ with respect to the class label (Breiman et al., 2017) and demographic parity $Gini_{\text{DP}}$ with respect to the sensitive attribute $a$ (Raff et al., 2018) as:

$$Gini_{\text{Accuracy}}(X) = \frac{1 - \sum_{c=1}^{|C|} \left( \frac{|X^c|}{|X|} \right)^2}{1 - \frac{1}{|C|}}, \quad Gini_{\text{DP}}(X) = \frac{1 - \sum_{s=1}^{|a|} \left( \frac{|X_s|}{|X|} \right)^2}{1 - \frac{1}{|a|}}, \tag{3}$$

in which $X^c$ and $X_s$ are training points with ground-truth label $c$ and sensitive attribute $s$, respectively.

**Definition 3** (Equalized Odds-aware Information Gain). *Given a prospective split over an attribute, we introduce the information gain with respect to the sensitive attribute conditioned on a class variable that we call equalized odds-aware information gain. The information gained over the split follows Equation 2 in which the equalized odds Gini index, $Gini_{Eodds}(X)$, is measured with respect to the sensitive attribute conditioned on class:*

$$Gini_{Eodds}(X) = \frac{|X^+|}{|X|} Gini_{Eodds}^+(X) + \frac{|X^-|}{|X|} Gini_{Eodds}^-(X),$$

$$Gini_{Eodds}^+(X) = \frac{1 - \sum_{s=1}^{|a|} \left( \frac{|X_s^+|}{|X^+|} \right)^2}{1 - \frac{1}{|a|}}, \quad Gini_{Eodds}^-(X) = \frac{1 - \sum_{s=1}^{|a|} \left( \frac{|X_s^-|}{|X^-|} \right)^2}{1 - \frac{1}{|a|}}. \tag{4}$$

$Gini_{\text{Eodds}}^+(X)$ and $Gini_{\text{Eodds}}^-(X)$ measure the impurity with respect to the sensitive attribute conditioned on the positive and negative class, respectively.

To find the best split among all possible splits, we propose a constrained optimization problem that maximizes the accuracy-based information gain and upper-bounds unfairness-based information gain:

$$\max Gain_{\text{Accuracy}}(\text{Split}) \quad \text{subject to } Gain_{\text{Unfairness}}(\text{Split}) \leq \tau. \tag{5}$$

$$\text{in which } Gain_{\text{Property}}(\text{Split}) = Gini_{\text{Property}}(\text{Parent}) - \sum_{k=1}^{2} \frac{|\text{Child}_k|}{|\text{Parent}|} Gini_{\text{Property}}(\text{Child}_k), \tag{6}$$

for Property $\in$ {Unfairness, Accuracy} in which Unfairness denotes a measure of unfairness such as demographic parity or equalized odds (see Definition 3). Algorithm 1 describes our fair tree training that supports different fairness metrics and can be verified efficiently.

---

**Algorithm 2:** ZK proof of demographic parity fair tree training. For equalized odds fair tree training see Appendix F.

---

**Input:** Training set $X$, Trained decision tree DT, Threshold $\tau$ over unfairness-based information gain.
**Output:** Commitment to trained decision tree parameters, ZK proof that parameters are fair.

1: Prover commits to the training data set and the trained decision tree, obtaining $[\![X]\!] = \{[\![x]\!]\}_{x \in X}$ and $[\![DT]\!]$.
2: Prover commits to two counting trees $[\![c^1]\!]$ and $[\![c^2]\!]$ initialized to all zeros.
   ▷ Phase 1:  ZK proofs of frequency
3: **for all** $x \in X$ **do**
4:     Prover locally computes the path that $x$ takes through DT, acquires the corresponding node indices, and commits to them, namely $\{[\![I_1]\!], \ldots, [\![I_h]\!]\}$.
5:     $[\![s]\!] \leftarrow [\![x[a]]\!]$.                                          ▷ Using ZK RAM access
6:     **for all** $j \in [1, h]$ **do**
7:         $[\![b]\!] \leftarrow ([\![x[DT[I_j].\mathsf{attr}]]\!] < [\![DT[I_j].\mathsf{val}]\!])$         ▷ $b = 0$ if $x$ goes to left child, $1$ otherwise
8:         Prover proves $([\![I_{j+1}]\!] = 2[\![I_j]\!] + [\![b]\!])$                    ▷ Prove that path is correct
9:         Prover updates commitments $[\![c^1[I_j]]\!] \leftarrow [\![c^1[I_j]]\!] + [\![s]\!]$ and $[\![c^2[I_j]]\!] \leftarrow [\![c^2[I_j]]\!] + [\![\neg s]\!]$
   ▷ Phase 2:  ZK proofs of fairness metric
10: **for all** $i \in \{\text{interior node indices}\}$ **do**
11:     Prover proves that:

$$\frac{[\![c^1[i]]\!] \times [\![c^2[i]]\!]}{([\![c^1[i]]\!] + [\![c^2[i]]\!])^2} - \frac{1}{[\![c^1[i]]\!] + [\![c^2[i]]\!]} \cdot \left( \sum_{b \in \{0,1\}} \frac{[\![c^1[2i+b]]\!] \times [\![c^2[2i+b]]\!]}{[\![c^1[2i+b]]\!] + [\![c^2[2i+b]]\!]} \right) \leq \tau/4$$

---

## 3.2 Efficient ZKP of fair training

In this section, we describe our ZK protocol which efficiently checks the fair training of the model. Note that in this paper we do not invent new cryptographic protocols to prove arbitrary programs $P$ (described in Section 2). Instead, our focus is to design a $P$ that verifies that a decision tree was trained fairly, and can be executed efficiently by a RAM model ZK protocol. To this end, in Section 3.1, we separated accuracy maximization and fairness requirements by modifying the split finding criterion. This means that one need only verify that the split point in the trained decision tree satisfies the fairness requirement, as opposed to exhaustively trying all possible split points. As a result, our ZK verification of fair training has a computational complexity lower than the training itself.

**Decision Tree and Commitment Representation.** Each internal node $n$ in a decision tree is represented as a tuple $(\mathsf{attr}, \mathsf{val})$, which respectively encode the attribute and threshold value by which $n$ splits the data. The decision tree DT of height $h$ is represented as an array of nodes, such that the two children of node $DT[i]$ are $DT[2i]$ and $DT[2i + 1]$. To avoid leaking information about the topology of the tree, the array will always have space for a full binary tree. Non-full trees can trivially be encoded using dummy threshold values to ensure that data never enters the missing nodes. We will use $[\![x]\!]$ to represent the cryptographic commitment of a value $x$ known to the prover. This means the prover can prove facts about $x$ without disclosing its value to the verifier.

**Our ZKP protocol.** Our efficient fairness verification is presented in Algorithm 2. It has two phases:

1. **Phase 1: ZK proofs of frequency counting.** This step takes the trained decision tree and the training data and, for each node and each possible sensitive attribute value, obtains committed counters on the number of training data entries falling into that node.
2. **Phase 2: ZK fairness metric verification.** Based on the committed counters, prove in ZK that the calculated fairness metrics from the counters are below the required fairness threshold.

Below we formally state the security of our protocol. Intuitively, the protocol does not reveal anything about the training dataset or the decision tree except to indicate that the fairness metric used for splitting is satisfied. Because a generic ZK protocol is used, the committed decision tree and the dataset can potentially be used to prove other statements. Below we state our main theorem in the security of the protocol, which is formally proven in Appendix E.

**Theorem 1.** *Algorithm 2 is a secure ZK proof of fair training.*

**Extensions.** In the above description, we show a ZK proof of fair training w.r.t. demographic parity. We note that it is easily generalizable to other fairness metrics mentioned in Section 3.1 as we can

|  | COMPAS | | Communities and Crime | | Default Credit | | Adult | |
|---|---|---|---|---|---|---|---|---|
|  | Eodds | Dem. Parity | Eodds | Dem. Parity | Eodds | Dem. Parity | Eodds | Dem. Parity |
| Running time | 12.74s | 9.87s | 8.31s | 7.37s | 72.21s | 50.80s | 104.73s | 62.86s |
| Communication | 28.7MB | 21.1MB | 23.9MB | 16.9MB | 107.3MB | 67.7MB | 145.2MB | 89.1MB |

Table 1: Efficiency of Confidential-PROFITT in terms of running time and communication costs on real-world datasets. Confidential-PROFITT executes the confidential proof of fair tree training in a matter of seconds with low communication overhead.

represent many group fairness metrics with a unified function such as:

$$\alpha T_a^+ + \beta F_a^+ = \alpha T_b^+ + \beta F_b^+$$

Demographic parity : $\alpha = \beta = 1$, Equalized odds : $\{\alpha = 1, \beta = 0\}$ and $\{\alpha = 0, \beta = 1\}$. (7)

For example, to support equalized odds, one must maintain four counter trees (rather than just $c^1$ and $c^2$, as in demographic parity) and verify the fairness metric accordingly. See our proposed ZK protocol for equalized odds in Appendix F.

## 4 EXPERIMENTAL EVALUATION

Our principal motivation for introducing Confidential-PROFITT is to prove to interested parties (e.g., users or auditor) that the learning algorithm used to train a model is fair by design—while protecting the confidentiality of both the training data and model. Next, we empirically evaluate the novel aspects introduced by Confidential-PROFITT: namely, (1) the ZK-friendly fairness-aware training algorithm along with (2) its customized ZK proof protocols. Therefore, we validate empirically the performance of Confidential-PROFITT as follows: i) **effectiveness in training a fair model**: The unfairness of decision trees decreases (while keeping the accuracy high) as the bound on the information gain w.r.t the sensitive attribute decreases; ii) **efficiency in proving the use of fair algorithm**: We implement our customized ZK protocol for verifying the algorithm and benchmark its runtime and communication costs. Our framework scales to real-world use cases (e.g., to a large number of training samples and attributes as well as deep decision trees).

We assess the performance of Confidential-PROFITT using four common datasets for fairness benchmarking: COMPAS (Angwin et al., 2016), Communities and Crime (Redmond, 2009), Adult Income (Adu, 1996), and Default Credit (Def, 2016). Refer to Appendix G.1 for details on these datasets. Two distinct code bases are utilized. We use the EMP-toolkit (Wang et al., 2016) to efficiently implement our ZK protocol and a JAVA and Python implementation to train and assess the accuracy and fairness of decision trees. Additional details about our implementations are in Appendix G.2.

**Effectiveness of Confidential-PROFITT.** Figure 1 shows the relationship between accuracy and fairness for decision trees with demographic parity and equalized odds fairness. These figures were constructed with a sweep of threshold values. We indicate the fairness and accuracy of trees trained without the fairness constraint as "Original" in Figure 1. We also compare Confidential-PROFITT to a "Baseline" for the demographic parity fair training method in Kamiran et al. (2010) and Raff et al. (2018). Note that this baseline is not applicable to trees trained for equalized odds fairness. Also note that the tree heights used for each dataset are described in Appendix G.1. The figures indicate that Confidential-PROFITT leads to increased fairness (shown by small fairness gap values) with only marginal, or sometimes no, decreases in accuracy compared to trees trained without fairness or with the baseline demographic parity fairness from Kamiran et al. (2010) and Raff et al. (2018). Interestingly, for some datasets, the range of accuracy values achieved exceeds the performance of trees trained without the fairness constraint. The accuracy drops themselves are often less than 10%. For example, for the Communities and Crime dataset we increase the fairness by almost 50% with at most a 5% accuracy drop. Therefore, **Confidential-PROFITT effectively improves the fairness.**

**Efficiency of Confidential-PROFITT.** Table 1 shows the runtime and communication costs of our fair training ZK protocols on Amazon EC2 machines in which the prover (i.e., the company) and the verifier (i.e., the auditor) are connected over a local area network (LAN). The runtime for all datasets is in the order of seconds. For example, it takes less than 10 seconds to prove the fair training in terms of demographic parity on COMPAS dataset. The communication overhead for both parties is at most 145 MB. For example, the total communication cost of running Confidential-PROFITT using equalized odds and demographic parity on the COMPAS dataset is 28 MB and 21 MB, respectively. Therefore, **Confidential-PROFITT efficiently proves the fair training of trees**.

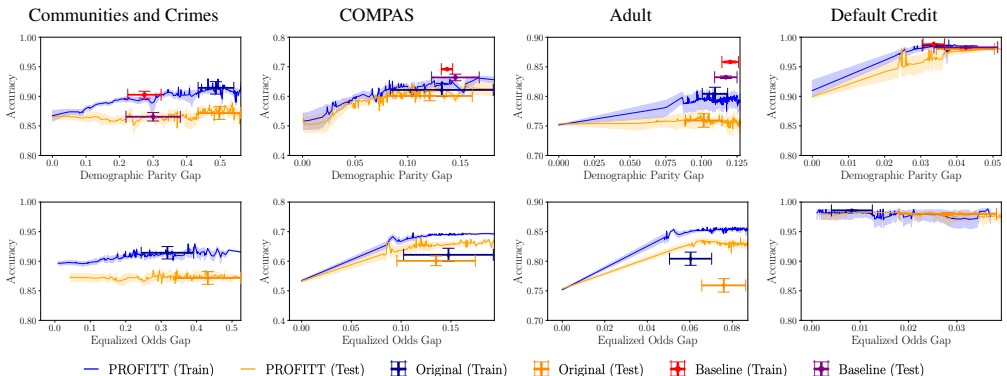

Figure 1: Effectiveness of Confidential-PROFITT in terms of demographic parity (first row) and equalized odds (second row) of trees. Confidential-PROFITT decreases the unfairness of the decision tree while keeping the accuracy close to that of the original decision tree trained without any fairness constraints. Original indicates a decision tree trained without any fairness constraint.

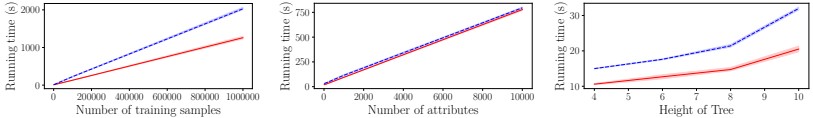

Figure 2: Scalability of Confidential-PROFITT using demographic parity and equalized odds. Confidential-PROFITT scales to a large number of training samples, a large number of attributes per sample, and large decision trees in practical running time (less than an hour). Default settings are: 10, 000 for training samples, 10 for attributes and 10 for tree height.

We further evaluate the scalability of Confidential-PROFITT by considering various numbers of training samples, number of attributes, and height of trees, which are $[10^3, 10^6]$, $[10, 10^4]$ and $[4, 10]$, respectively. Figure 2 shows the effect of the number of training samples, number of attributes and height of trees on the running time (see Appendix H for the communication cost) of Confidential-PROFITT when proving demographic parity and equalized odds. These results demonstrate that even for the maximum values of these parameters, the running time is less than an hour. Therefore, **Confidential-PROFITT is scalable**. Next, we discuss the pattern of running time.

As expected, we observe linear scaling with the number of samples (left panel). The reason for this is that each sample is processed using a constant number of operations during the first phase of our protocol, and the number of operations in the second phase is unchanged by the number of samples. The equalized odds protocol incurs higher running times, since it has to initialize and update four counting trees while the demographic parity version only needs to keep track of two. We also observe that the protocol scales linearly with the number of attributes (middle panel). This is because increasing the number of attributes increases the number of operations required to commit to the dataset. We note that the demographic parity and equalized odds protocols are roughly equivalent in run time for these experiments because committing to the dataset takes the same amount of work in both versions. Finally, we observe the beginning of exponential scaling as tree height is increased (right panel). This is due to the exponentially increasing size of the trained decision tree and counting trees, as well as the fairness checking phase (which iterates through all interior nodes). Breakdown of computational costs between phases of the ZK protocol varied depending on parameters. For example, on the COMPAS dataset, the commitment, counting, and fairness checking phases took up approximately 60%, 32%, and 0.2% of the runtime respectively whereas for the Default Credit dataset, those respective values became 24%, 72% and 2% (the remainder of the runtime is spent initializing the underlying ZK framework). This difference is caused by the size difference between the two datasets.

**An analysis on fair splitting criterion versus fairness metrics.** Figure 3 demonstrates a connection between the fair learning algorithm and the fairness metrics investigated: the unfairness of demographic parity and equalized odds increases with the average unfairness gain values over all node splits for decision trees. This is intuitive given that similarly to the label-wise discrimination encouraged by $Gain_{\text{Accuracy}}$, high values of $Gain_{\text{Unfairness}}$ may bring the tree to discriminate across values of the sensitive attribute. By limiting this discrimination, we expect little change to the purity of successive nodes, eventually leading to leaves having a mixed population from each subgroup.

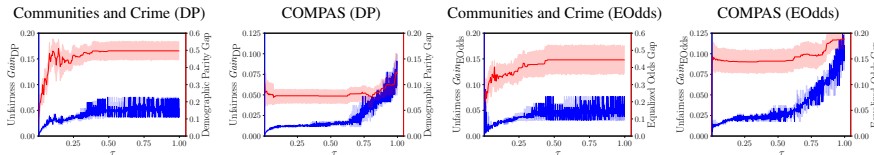

Figure 3: Relationship between fair splitting criteria and unfairness definitions. As the bound on the unfairness-based information gain decreases , the unfairness of trained decision trees decreases.

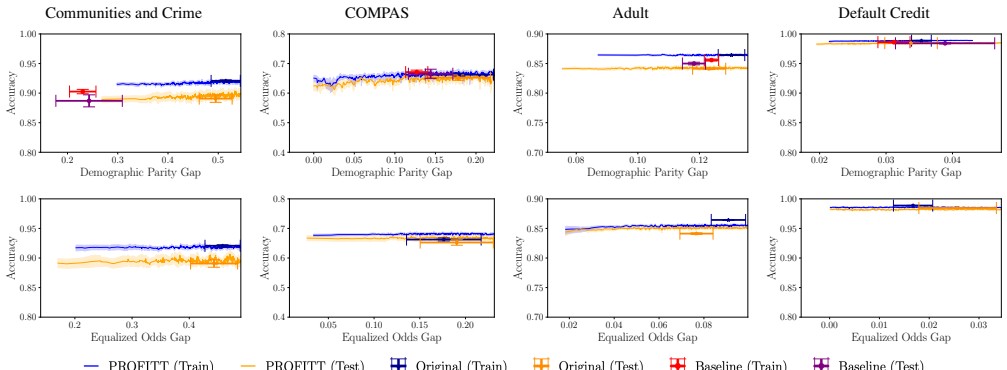

Figure 4: Effectiveness of Confidential-PROFITT in terms of demographic parity (first row) and equalized odds (second row) of random forests. Confidential-PROFITT decreases the unfairness of the random forest while keeping the accuracy close that of the original random forest trained without fairness constraints. Original indicates random forest trained without any fairness constraint.

| COMPAS | | Communities and Crime | | Default Credit | | Adult | |
|---|---|---|---|---|---|---|---|
| Eodds | Dem. Parity | Eodds | Dem. Parity | Eodds | Dem. Parity | Eodds | Dem. Parity |
| 108.73s | 73.79s | 73.49s | 51.88s | 652.9s | 372.55s | 908.63s | 537.59s |

Table 2: Confidential-PROFITT executes the proof of fair random forests training in practical time.

## 5 CONFIDENTIAL PROOF OF FAIR TRAINING OF ENSEMBLES OF TREES

Ensembling decision trees into a random forest can improve the performance of the resulting classifier (see Appendix I for these results). Therefore, we extend Confidential-PROFITT to prove the fairness of random forests (Zhang et al., 2020b; 2021; Raff et al., 2018). Concretely, we construct fair random forests from ensembles of fair decision trees with bagging and random feature selection (see Appendix G.2 for details of how to train random forests). By running a secure coin-flipping protocol between the company and the auditor before training, the randomness used for subsampling data points and attributes during the ensemble process can be revealed to the auditor. Given this, an efficient protocol for ZK proof of fairness for a random forest can be derived by performing ZK proofs of each decision tree and data consistency between different trees. See details in Appendix J. Figure 4 and Table 2 show accuracy-fairness relationships and running time of random forests. We observe similar results: **Confidential-PROFITT achieves major fairness gains for random forests with marginal changes to the accuracy in a practical time**.

## 6 CONCLUSION

In this paper, we proposed a framework that can be used to confidentially prove the fair training of a decision tree. Confidential-PROFITT can help companies avoid receiving fairness-related penalties as it certifies the fairness during training prior to making any decisions on clients' queries.

**Practical considerations.** Across the whole data-based decision process, Confidential-PROFITT can prove the fairness of the model-training phase. To provide an end-to-end fairness guarantee to the auditor, one could extend our framework by proving the fairness of the data source and by proving the inference-time use of the model. The company can use fairness-aware data pre-processing to ensure a fair representation of the different subgroups (e.g., by swapping the ground-truth labels (Kamiran & Calders, 2009; Luong et al., 2011)) and prove the integrity of this process by providing a ZK proof to the auditor. To ensure the company uses the proven fair model in the deployment phase, one could use existing ZK-based verified inferences for trees (Zhang et al., 2020a; Singh et al., 2021).

## ACKNOWLEDGMENTS

We would like to acknowledge our sponsors, who support our research with financial and in-kind contributions: CIFAR through the Canada CIFAR AI Chair program and the Catalyst grant program, Microsoft, and NSERC through the Discovery Grant and COHESA Strategic Alliance. Resources used in preparing this research were provided, in part, by the Province of Ontario, the Government of Canada through CIFAR, and companies sponsoring the Vector Institute. We would like to thank members of the CleverHans Lab for their feedback. Sébastien Gambs is supported by the Canada Research Chair program, a Discovery Grant from NSERC as well as the NSERC-RDC DEEL project. Xiao Wang is supported by DARPA under Contract No. HR001120C0087, NSF award #2016240, and research awards from Meta and Google. Adrian Weller acknowledges support from EPSRC grant EP/V056883/1, a Turing AI Fellowship under EP/V025279/1, and the Leverhulme Trust via CFI.

## ETHICS STATEMENT

The main objective behind Confidential-PROFITT is to achieve provably fair training without the requirements of revealing the training data or model to an auditor. In particular, the combination of confidentiality and fairness alleviates companies' data and intellectual property concerns for auditing, which could increase fairness auditing and public trust in audited AI overall.

The main ethical issues of this work are related to the definitions of fairness and benchmark datasets used. Indeed, algorithmic fairness is composed of multitudes of conflicting definitions. Our work focuses on group fairness, and specifically on demographic parity and equalized odds, both of which are popular and relevant metrics. However by relying on these definitions, we acknowledge that this reinforces the political and philosophical worldviews and ideologies underlying them, and deprives other notions of fairness (e.g., individual fairness or settings in which there is no given sensitive attribute) of due attention.

The datasets used to validate our methods are popular fairness benchmarks. Unfortunately, their popularity can be partially attributed to the blatant unfairness of the models they have produced and the harm these models may have done when deployed in practice. Our use of these benchmark datasets is merely to remain comparable with other approaches in the literature—we do not necessarily support the use of machine learning in any of these contexts, specifically in recidivism prediction (COMPAS) or community crime profiling (Communities and Crime).

To account for these risks, we advocate for more research at the intersection of privacy and fairness, particularly for other definitions of fairness. We also believe that it is important to foster more research on the development of practical solutions in fairness and fairness auditing.

## REPRODUCIBILITY STATEMENT

To support our main contributions, we have provided a detailed report of our algorithms and proofs in appendices. More precisely, we have detailed our datasets, hyperparameters and implementations in Appendix G.1 and G.2. The code is available at `https://github.com/cleverhans-lab/Confidential-PROFITT`. This work is licensed under a Creative Commons Attribution-NonCommercial 4.0 International License `https://creativecommons.org/licenses/by-nc/4.0/`.

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

# A  NOTATION

Table 3 shows the notation used throughout this paper.

Table 3: Notation table.

| Notation | Meaning | Notation | Meaning |
|---|---|---|---|
| $X$ | Dataset | $Y$ | Set of binary ground-truth labels |
| $a$ | Binary sensitive attribute | $s$ | Sensitive attribute value |
| $X_s$ | Dataset with sensitive attribute $s$ | $X^c$ | Dataset with ground-truth label $c$ |
| DT | Decision tree | $h$ | Decision tree height |
| $\tau$ | Threshold | $A(\cdot, \cdot)$ | Fairness-aware training algorithm |
| $\hat{Y}$ | Set of binary predicted labels | *Gain* () | Information gain |
| *Gini* () | Gini | $[\![x]\!]$ | Commitment of $x$ |
| $\Pi$ | Zero-knowledge proof protocol | $c$ | Counter |

# B  FAIRNESS METRICS

In this section, we describe four common fairness metrics: *Demographic Parity*, *Equalized Odds*, *Equal Opportunity*, and *Predictive Equality*. We consider the former two metrics in this paper, though our work may be easily extended to these other metrics as equal opportunity and predictive equality are weaker versions of equalized odds.

**Definition 4** (Demographic parity (Calders et al., 2009)). *A predictor $\hat{Y}$ satisfies Demographic Parity with respect to the sensitive attribute $S$ if:*

$$\Pr[\hat{Y} = 1 | S = a] = \Pr[\hat{Y} = 1 | S = b] \qquad \forall a, b \in S.$$

Satisfying demographic parity, or statistical group parity, refers to equal probability of a predicted positive label across subgroups partitioned by protected attribute $S$. In essence, enforcing demographic parity ensures that the predicted class under the model is independent of inclusion in a particular subgroup. Such a fairness definition leads to situations and tasks in which the prediction is intended to be independent of subgroup inclusion. Additionally, demographic parity presents a fairness metric that can, to some extent, overcome issues of bias in the true label associated with $S$, as demographic parity ignores the true label in its formulation. Although several tasks necessitate such properties in a fairness metric, classification in domains in which target labels may be intrinsically related to subgroup values, such as healthcare, demographic parity fails to capture key relationships between target labels and subgroups that can lead to poorer fairness (in the societal sense of fairness). For example, prediction in breast cancer diagnosis needs to incorporate subgroup information about biological sex, due to higher rates in women.

**Definition 5** (Equalized Odds (Hardt et al., 2016)). *A predictor $\hat{Y}$ satisfies Equalized Odds with respect to the sensitive attribute $S$ if:*

$$\Pr[\hat{Y} = 1 | Y = y, S = a] = \Pr[\hat{Y} = 1 | Y = y, S = b.] \qquad \forall y \in \{0, 1\}, \forall a, b \in S$$

The equalized odds metric is a strict constraint on fairness that requires a predictor to jointly equalize the false positive rate (FPR) and true positive rate (TPR) across subgroups determined by $S$. In the equalized odds formulation, the fairness of a model depends not only on the predicted label but on the true label distribution as well. Satisfying equalized odds ensures that the predicted class is conditionally independent of the protected attribute and the target class. Equalized odds was proposed to alleviate pitfalls of demographic parity, which cannot capture important relationships between the target class and the subgroup's information in its formulation of fairness. Therefore, equalized odds is better suited for tasks in which intrinsic information links subgroup inclusion and true class, but can propagate bias in cases where the "true" label encodes societal unfairness. For equalized odds fairness, the prediction should be completely independent of the sensitive attribute. Note that equalizing both FPR and TPR implies searching for the intersection of per-subgroup

area-under-receiver-operator-curves (AUROC), which may not be always satisfiable for non-trivial intersections. Similarly, such an intersection may lie in an undesirable region of the parameter space that significantly degrades the utility of the model overall. Thus, relaxation and calibration methods have spawned in order to mitigate these issues with this fairness metric (Hardt et al., 2016).

**Definition 6** (Equal Opportunity (Hardt et al., 2016)). *A predictor $\hat{Y}$ satisfies Equal Opportunity with respect to the sensitive attribute $S$ if:*

$$\Pr[\hat{Y} = 1 | Y = 1, S = a] = \Pr[\hat{Y} = 1 | Y = 1, S = b] \qquad \forall a, b \in S.$$

Equal opportunity introduces a weaker version of equalized odds, in which solely TPR is equalized across subgroups. Due to the relative difficulties associated with satisfying equalized odds, equality of opportunity relaxes the metric by permitting unequalized FPR, but continues to require TPR, and offers an advantage over demographic parity in certain settings by incorporating the target label into its formulation. Equality of opportunity largely works in domains in which the positive class could be viewed as "desirable" (thus the name *opportunity*) by enforcing equal inclusion in the positive class conditionally independent of $S$ and *true* inclusion in the positive class.

**Definition 7** (Predictive Equality (Chouldechova, 2016; Corbett-Davies et al., 2017)). *A predictor $\hat{Y}$ satisfies Predictive Equality with respect to the sensitive attribute $S$ if:*

$$\Pr[\hat{Y} = 1 | Y = 0, S = a] = \Pr[\hat{Y} = 1 | Y = 0, S = b] \qquad \forall a, b \in S.$$

Similar to equal opportunity, predictive equality enforces a partial form of equalized odds requiring equalized FPR across subgroups with no constraints on TPR. Conversely to equal opportunity, predictive equality applies well in scenarios in which the inclusion in the positive class under the model can be considered a poor outcome, and therefore causes considerable damage when the *true* label is negative.

Equality of opportunity and predictive equality both can be seen as partial requirements for equalized odds:

$$\begin{aligned} &\alpha T_a^+ + \beta F_a^+ = \alpha T_b^+ + \beta F_b^+ \\ &\text{Equalized odds}: \{\alpha = 1, \beta = 0\} \text{ and } \{\alpha = 0, \beta = 1\} \\ &\text{Equal opportunity}: \{\alpha = 1, \beta = 0\} \\ &\text{Predictive equality}: \{\alpha = 0, \beta = 1\}. \end{aligned} \qquad (8)$$

Therefore, any predictor satisfying equalized odds will satisfy both predictive equality and equal opportunity.

## C   BLOCK DIAGRAM OF CONFIDENTIAL-PROFITT

Figure 5 depicts an overview of Confidential-PROFITT which consists of four main steps.

## D   DECISION TREE TRAINING

### D.1   DECISION TREE TRAINING WITHOUT FAIRNESS CONSTRAINTS

Decision trees used for classification are trained recursively from root to leaf node by successively partitioning the input set, as shown in Algorithm 3. Ideally, data points in the leaf nodes are pure with respect to class and the leaves are used for prediction during inference. Partitioning is represented by nodes in a binary tree and enforced with a Boolean condition that splits the data based on some attribute value into at most two disjoint child nodes. To find the optimal splitting criterion for a node, the algorithm iterates over every attribute and attribute value. These values are used as thresholds to split the data into child nodes. The algorithm measures the information gained (with respect to node class purity) between the parent and child nodes and chooses the splitting criterion that maximized this gain. The process of finding the best split is detailed in Algorithm 4. Splitting continues until some maximum height is reached or until the node sizes have reached some minimum value. While there are many decision tree training algorithms, this paper uses the CART (Classification and Regression Tree) with information gain, Gini impurity, binary splitting, and binary classification for numerical and categorical datasets. The CART algorithm has been popular since its inception in 1983 (Breiman et al., 1983).

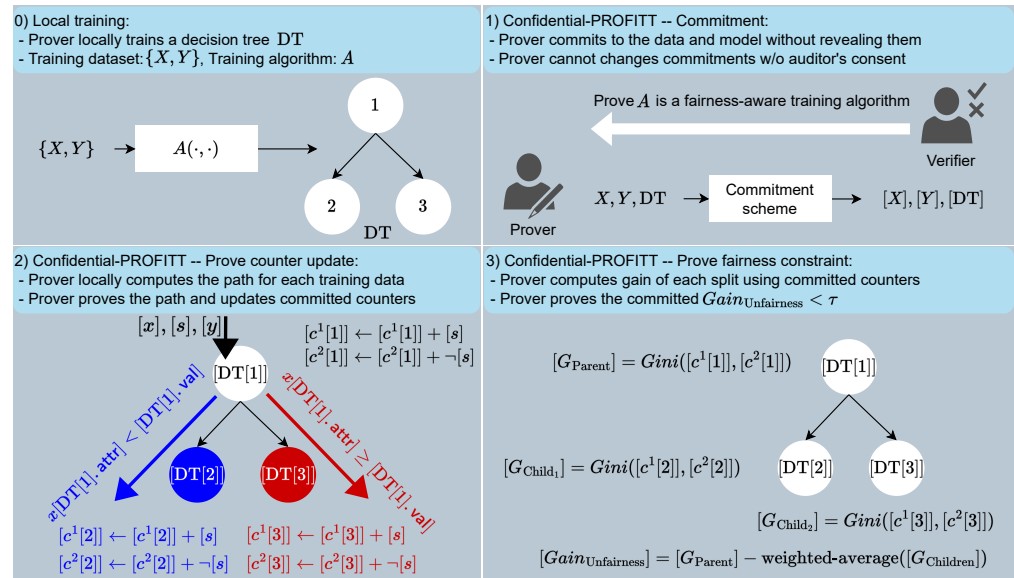

Figure 5: Block diagram of the proposed Confidential Proof of Fairness, Confidential-PROFITT.

---

**Algorithm 3:** Recursively building decision tree.

**Input:** Training dataset $X$, current decision tree height $h_c$, maximum decision tree height $h$

**Output:** Trained decision tree DT

1: **if** DT.height $> h$ **then**
2:     **return** null
3: split = FindBestSplit($X$)
4: **if** split.leafs $> 1$ **then**
5:     **for** leaf $\in$ split **do**
6:         **return** and link BuildTree(leaf.X, DT.height$+=1$, $h$)

---

**Algorithm 4:** Finding the best (fairness-oblivious) split for each node.

**Input:** Dataset $X$ in $j$-th node DT [j].

**Output:** Best split.

1: Accuracy_gains $= []$             ▷ Initialize gains for each value of each attribute
2: Parent $= X$             ▷ Make parent node
3: **for** attr $\in$ Attributes **do**
4:     **for** $x \in X$ **do**
5:         val $= x[\text{attr}]$          ▷ Set split threshold value per datapoint
6:         Child$_1 = \{x \in X | x[\text{attr}] < \text{val}\}$, Child$_2 = \{x \in X | x[\text{attr}] \geq \text{val}\}$      ▷ Make children
7:         $Gain_{\text{Accuracy}} = Gini_{\text{Accuracy}}(\text{Parent}) - \sum_{k=1}^{2} \frac{|\text{Child}_k|}{|\text{Parent}|} Gini_{\text{Accuracy}}(\text{Child}_k)$      ▷ Gain wrt label
8:         Accuracy_gains.append(attr, val, $Gain_{\text{Accuracy}}$)
9: **return** DT$[j]$.attr, DT$[j]$.val, DT$[j]$.gains $= \max_{Gain_{\text{Accuracy}}}$ Accuracy_gains      ▷ Best split

---

## D.2   DECISION TREE TRAINING WITH FAIRNESS CONSTRAINTS

Training a decision tree with fairness while maintaining the root-to-leaf recursive training procedure only requires one additional step, detailed in Algorithm 4. While iterating over the various splits, we measure both information gain with respect to class Gini impurity and fairness Gini impurity. We select the splits that produced fairness information gains below some threshold before choosing whichever had the highest class information gain. This leads to a high fairness and accuracy for the dataset.

---

**Functionality $\mathcal{F}_{ZKDT}$**

1. Prover and Verifier send $\tau$, abort if they do not match.
2. Prover sends dataset $X$, decision tree parameters DT
3. Verifier receives 1 if $\forall$ interior node indices $i$, $Gain_{\text{Unfairness}}(i, X, \text{DT}) \leq \tau$ holds. Verifier receives 0 otherwise.

---

Figure 6: Ideal functionality for verifying the fairness of a decision tree in zero knowledge .

---

**Functionality $\mathcal{F}_{ZK}$**

**ZK Operations**

- **Commitment to private inputs:** On receiving ($\texttt{Input}, x$) from $P$, store $x$ and send $[\![x]\!]$ to each party.
- **Commitment to (public) constants:** On receiving ($\texttt{Const}, x$) from both parties, store $x$ and send $[\![x]\!]$ to each party (if the two inputs do not match, both parties receive $\texttt{cheating}$).
- **Boolean circuit satisfiability:** On receiving ($\texttt{Circ}, C, [\![x_0]\!], \cdots, [\![x_{n-1}]\!]$) from both parties, in which the $x$'s are Boolean values and $C$ is a Boolean circuit, compute $b := C(x_0, \cdots, x_{n-1})$ and send $b$ to $V$.

**RAM Operations**

- **Initialization:** On receiving ($\texttt{Init}, N$) from $P$ and $V$, store an $N$-value array $A$ with values initialized to $\perp$, and set $f := \texttt{honest}$.
- **Write:** On receiving ($\texttt{Write}, [\![i]\!], [\![d]\!]$) from $P$ and $V$, set $A[i] := d$ and send fresh $[\![d]\!]$ to each party.
- **Read:** On receiving ($\texttt{Read}, [\![i]\!], d$), from $P$, ($\texttt{Read}, [\![i]\!]$) from $V$, send $[\![d]\!]$ to both parties. If $d \neq A[i]$ set $f := \texttt{cheating}$.
- **Check:** On receiving ($\texttt{Check}$) from $V$, send $f$ to both parties.

---

Figure 7: Ideal functionality for Zero Knowledge Proofs with RAM access.

## E  PROOF OF SECURITY OF THEOREM 1

We prove the property of our protocol based on the composition paradigm Canetti (2000), which is the standard way for proving cryptographic protocols. We represent the ideal functionality for our zero knowledge fairness check $\mathcal{F}_{ZKDT}$ in Figure 6. To realize this functionality, the protocol described in Algorithm 2 specifies a series of calls to a Zero-Knowledge Proof functionality with RAM capabilities $\mathcal{F}_{ZK}$, which is given in Figure 7. This can be instantiated using existing works that construct RAM-based zero-knowledge proofs.

*Proof.* A given execution of the protocol in Algorithm 2 comes with a prescribed pattern of calls to $\mathcal{F}_{ZK}$ – the low level details of this pattern are abstracted in the algorithm, but specified exactly in the code that implements our protocol. Any deviation from this pattern of calls to $\mathcal{F}_{ZK}$ is detected trivially (since the honest party sends a command to $\mathcal{F}_{ZK}$ that goes unanswered), and results in $\texttt{abort}$ of the protocol. Thus no adversary $\mathcal{A}$ is able to break security by deviating from this pattern. For those adversaries that follow the correct pattern of calls, we construct simulators as follows.

**Malicious Prover:** Simulator $S$ interacts with $\mathcal{F}_{ZKDT}$ with the following procedure, running malicious Prover $\mathcal{A}$ in the $\mathcal{F}_{ZK}$-hybrid model as a subroutine:

1. Simulating $\mathcal{F}_{ZK}$, $S$ receives ($\texttt{Init}, N$) from $\mathcal{A}$, in which $N$ is the size of RAM required to represent the computation. Store flag $f := \texttt{honest}$. Initialize an array $A$ in accordance with $\mathcal{F}_{ZK}$. Next $S$ receives a sequence of $\texttt{Input}$ commands from $\mathcal{A}$, followed by $\texttt{Const}$ commands to obtain RAM indices, and $\texttt{Write}$ commands to store the values in the RAM. Write the values into $A$ and interpret them as $X$ and DT. Send back commitments $[\![X]\!]$ and $[\![\text{DT}]\!]$.
2. Receive a sequence of ($\texttt{Const}, 0$) and ($\texttt{Const}, \text{RAM index}$) followed by $\texttt{Write}$ commands to initialize $c^1$ and $c^2$. Similarly to the previous step, store these values in $A$ and send back $[\![c^1]\!]$ and $[\![c^2]\!]$.
3. For all $x_i \in X$ do:
    4. Receive $\texttt{Input}$, $\texttt{Const}$, and $\texttt{Write}$ commands to store $\{I_1, \cdots, I_H\}$. Write the values in $A$, and send back $\{[\![I_1]\!], \cdots, [\![I_H]\!]\}$.
    5. Receive ($\texttt{Read}, [\![\text{index of } x[a]]\!], s$), if $A[\text{index of } x[a]] \neq s$, set $f := \texttt{cheating}$. Send back $[\![s]\!]$.
    6. For all $j \in [1, H]$ do:

7. Receive a bit $b$ via `Input` command, followed by a sequence of `Reads` to retrieve $x[\text{DT}[I_j].\texttt{attr}]$, $\text{DT}[I_j].\texttt{val}$, and a `Circ` command to verify that $b = x[\text{DT}[I_j].\texttt{attr}] < \text{DT}[I_j].\texttt{val}$. Send back values obtained by executing $\mathcal{F}_{ZK}$ faithfully.

8. Receive commands to verify $[\![I_{j+1}]\!] = 2[\![I_j]\!] + [\![b]\!]$, execute $\mathcal{F}_{ZK}$ faithfully.

9. Receive commands to verify updates $c^1[I_j] + s$ and $c^2[I_j] + \neg s$ and write the updates to the RAM. Execute $\mathcal{F}_{ZK}$ faithfully.

10. For all $i \in \{\text{interior node indices}\}$ do

 - Receive `Read` commands to retrieve $c^1[i], c^1[2i], c^1[2i+1], c^2[i], c^2[2i], c^2[2i+1]$, followed by `Circ` commands to carry out the fairness check. Execute $\mathcal{F}_{ZK}$ faithfully.

11. After protocol execution, execute `Check` command. If $f = \texttt{cheating}$, send $\perp$ to $\mathcal{F}_{ZKDT}$ and abort. Otherwise, send $X$ and DT to $\mathcal{F}_{ZKDT}$.

Clearly $S$ constructs a view for $\mathcal{A}$ that is indistinguishable from real world execution, since all interactions with $\mathcal{A}$ through $\mathcal{F}_{ZK}$ are performed faithfully. Input extraction follows trivially since $\mathcal{A}$ sends the values of $X$ and DT to $\mathcal{F}_{ZK}$ in the commitment phase. Further, we have that the generated view is consistent with the outputs generated by $\mathcal{A}$'s inputs, since the fairness checking phase directly computes (an algebraic rearrangement of) $Gain_{\text{Unfairness}}(i, X, \text{DT}) \leq \tau$ for each interior node index $i$.

**Malicious Verifier:** Simulator $S$ interacts with $\mathcal{F}_{ZKDT}$ with the following procedure, running malicious Verifier $\mathcal{A}$ in the $\mathcal{F}_{ZK}$-hybrid model as a subroutine:

1. Simulating $\mathcal{F}_{ZK}$, $S$ receives $(\texttt{Init}, N)$ from $\mathcal{A}$, where $N$ is the size of RAM required to represent the computation. Store flag $f := \texttt{honest}$. Initialize an array $A$ in accordance with $\mathcal{F}_{ZK}$. Next, simulate a series of `Input` commands by sending $[\![0]\!]$ to $\mathcal{A}$ several times (with fresh randomness each time). Then receive a series of `Const` commands to obtain commitments to RAM indices – simulate $\mathcal{F}_{ZK}$ by sending back commitments to the requested indices. Next receive `Write` commands from $\mathcal{A}$ for storage of the dataset in the RAM. Store 0 at each requested index and send $[\![0]\!]$ to simulate responses to the `Write` commands. In this way, $\mathcal{A}$ receives a list of commitments that stand in for $[\![X]\!]$ and $[\![\text{DT}]\!]$.

2. Receive a sequence of $(\texttt{Const}, 0)$ and $(\texttt{Const}, \text{RAM index})$ followed by `Write` commands to initialize $c^1$ and $c^2$. Give commitments to the requested `Const` values faithfully, store these values in $A$, and send back $[\![c^1]\!]$ and $[\![c^2]\!]$.

3. For all $x_i \in X$ do:

4. To simulate commitment to $\{I_1, \cdots, I_H\}$, send $[\![0]\!]$ for `Input` commands, and faithful commitments to RAM indices when `Const` commands are received from $\mathcal{A}$. Simulate `Write` commands by storing 0 at the requested indices in $A$, and send $[\![0]\!]$ to stand in for the commitments $\{[\![I_1]\!], \cdots, [\![I_H]\!]\}$.

5. Receive $(\texttt{Read}, [\![\text{index of } x[a]]\!])$, simulate honest execution of $\mathcal{F}_{ZK}$ by sending back $[\![0]\!]$ and keeping $f = \texttt{honest}$.

6. For all $j \in [1, H]$ do:

7. Send $[\![b]\!]$ where $b = 0$. Receive `Read` commands to retrieve $x[\text{DT}[I_j].\texttt{attr}]$ and $x[\text{DT}[I_j].\texttt{val}]$ – simulate by sending $[\![0]\!]$ and keeping $f = \texttt{honest}$. Next receive a `Circ` command to verify that $b = x[\text{DT}[I_j].\texttt{attr}] < \text{DT}[I_j].\texttt{val}$ – send back 1 indicating that the equality holds.

8. Receive commands to verify $[\![I_{j+1}]\!] = 2[\![I_j]\!] + [\![b]\!]$, as in previous steps send $[\![0]\!]$ to simulate commitments for the `Read` commands (while keeping $f = \texttt{honest}$) and send 1 in response to `Circ` commands indicating that the desired equality holds.

9. Receive commands to verify updates $c^1[I_j] + s$ and $c^2[I_j] + \neg s$ and write the updates to the RAM. Simulate the verification as in the previous two steps, and send $[\![0]\!]$ in response to the `Write` commands.

10. For all $i \in \{\text{interior node indices}\}$ do

 - Receive `Read` commands to retrieve $c^1[i], c^1[2i], c^1[2i+1], c^2[i], c^2[2i], c^2[2i+1]$, followed by `Circ` commands to carry out the fairness check. As before, send $[\![0]\!]$ to simulate commitments and send 1 to indicate that the fairness checks pass.

11. After protocol execution, receive `Check` command. Simulate $\mathcal{F}_{ZK}$ by sending `honest` to $\mathcal{A}$.

---

**Algorithm 5:** Zero-knowledge proof of equalized odds-aware tree training.

---

**Input:** Training set $X$, trained decision tree DT, $Gain_{\text{Unfairness}}$ threshold $\tau$
. **Output:**

1: Prover commits to the training data set and the trained decision tree, obtaining $[\![X]\!] = \{[\![x]\!]\}_{x \in X}$ and $[\![\text{DT}]\!]$.
2: Prover commits to four counting trees $[\![c^1]\!]$, $[\![c^2]\!]$, $[\![c^3]\!]$, and $[\![c^4]\!]$ initialized to all zeros.

   ▷ Phase 1: ZK proofs of frequency
3: **for all** $x \in X$ **do**
4:    Prover locally computes the path that $x$ takes through DT, acquires the corresponding node indices, and commits to them, namely $\{[\![I_1]\!], \ldots, [\![I_H]\!]\}$.
5:    $[\![s]\!] \leftarrow [\![x[a]]\!]$.                                          ▷ Using ZK RAM access
6:    $[\![y]\!] \leftarrow [\![x[Y]]\!]$.
7:    $[\![u_1]\!] \leftarrow [\![y]\!] \wedge [\![s]\!]$
8:    $[\![u_2]\!] \leftarrow [\![y]\!] \wedge [\![\neg s]\!]$
9:    $[\![u_3]\!] \leftarrow [\![\neg y]\!] \wedge [\![s]\!]$
10:    $[\![u_4]\!] \leftarrow [\![\neg y]\!] \wedge [\![\neg s]\!]$                              ▷ counting tree updates
11:    **for all** $j \in [1, H]$ **do**
12:        $[\![b]\!] \leftarrow ([\![x[\text{DT}[I_j].\text{attr}]]\!] < [\![\text{DT}[I_j].\text{val}]\!])$     ▷ $b = 0$ if $x$ goes to left child, 1 otherwise
13:        Prover proves $([\![I_{j+1}]\!] = 2[\![I_j]\!] + [\![b]\!])$         ▷ Prove that path is correct
14:        Prover updates commitments $[\![c^k[I_j]]\!] \leftarrow [\![c^k[I_j]]\!] + [\![u_k]\!]$ for $k \in \{1, 2, 3, 4\}$.

   ▷ Phase 2: ZK proofs of fairness metric
15: **for all** $i \in \{\text{interior node indices}\}$ **do**
16:    Prover proves that:

$$\frac{[\![c^1[i]]\!] \times [\![c^2[i]]\!]}{([\![c^1[i]]\!] + [\![c^2[i]]\!])^2} - \frac{1}{[\![c^1[i]]\!] + [\![c^2[i]]\!]} \cdot \left( \sum_{b \in \{0,1\}} \frac{[\![c^1[2i+b]]\!] \times [\![c^2[2i+b]]\!]}{[\![c^1[2i+b]]\!] + [\![c^2[2i+b]]\!]} \right) \leq \tau/4$$

17:    Prover additionally proves that:

$$\frac{[\![c^3[i]]\!] \times [\![c^4[i]]\!]}{([\![c^3[i]]\!] + [\![c^4[i]]\!])^2} - \frac{1}{[\![c^3[i]]\!] + [\![c^4[i]]\!]} \cdot \left( \sum_{b \in \{0,1\}} \frac{[\![c^3[2i+b]]\!] \times [\![c^4[2i+b]]\!]}{[\![c^3[2i+b]]\!] + [\![c^4[2i+b]]\!]} \right) \leq \tau/4$$

---

$S$ constructs a view for $\mathcal{A}$ that is indistinguishable from real world execution, since all handles $[\![0]\!]$ given to $\mathcal{A}$ are indistinguishable from those during real-world execution, and because in simulating $\mathcal{F}_{ZK}$, $S$ can trivially indicate that the component circuits of the protocol are satisfied properly. □

We would like to highlight that our approach relies on zero-knowledge proofs to formally guarantee the confidentiality of the learned model as well as of the data even against a malicious auditor (see the above mathematical proof). To be more precise, the confidentiality of the model and data is ensured by the "zero-knowledge" property of the underlying cryptographic protocol that can tolerate any malicious behavior (Goldwasser et al., 1985; Goldreich et al., 1991). The security of these protocols are in turn based on standard hardness assumptions used to secure the Internet. We note that the zero-knowledge proof protocol itself is not the focus of this paper; instead, we show how we can use existing zero-knowledge proof protocols to prove fairness-aware training in a smart way for high efficiency. The confidentiality of our system can be ensured by any secure zero-knowledge proof system (which is what Appendix E proves), including the one used in this paper or others systems.

## F   OUR ZERO-KNOWLEDGE PROOF PROTOCOL FOR TREE TRAINING WITH EQUALIZED ODDS FAIRNESS

Algorithm 5 follows the same ideas as Algorithm 2, except it proves that $Gain^+_{Eodds}$ and $Gain^-_{Eodds}$ do not exceed the threshold $\tau$, rather than $Gain_{DP}$. As in Algorithm 2, these metrics are computed and proved by counting the number of samples of particular categories that pass through each node in the tree. However, to compute $Gain^+_{Eodds}$ and $Gain^-_{Eodds}$ we need to keep track of *four* categories of samples rather than just two: samples that are in the protected class with true positive label, samples in the unprotected class with true positive label, samples in the protected class with true negative label, and samples in the unprotected class with true negative label. To keep track of these quantities at each node, we use four counting trees, whose commitments are updated by adding Boolean values $u_1$ through $u_4$ indexing which category a given sample falls into. In Phase 2 of the algorithm, the sample counts are used to show that in each interior node, the $Gain^+_{Eodds}$ and $Gain^-_{Eodds}$ do not exceed the threshold $\tau$.

| Dataset | #Samples | #Attr. | Sensitive attribute | Task | Tree height |
|---|---|---|---|---|---|
| COMPAS | 6,151 | 8 | Race (Binarized) | Yes/No recidivism | 6 |
| Crime | 1,993 | 22 | Race % Black $\geq$ .06 | Crime rate $\geq$ .7 | 4 |
| Default Credit | 30,000 | 23 | Age $\geq$ 25 | Good/Bad credit | 10 |
| Adult | 45,222 | 14 | Gender (Binarized) | Income $\geq 50k$ | 10 |

Table 4: Summary of datasets.

# G  DETAILS ON EXPERIMENTAL SETUP

## G.1  DETAILS ON DATASETS

We consider four common datasets for fairness benchmarking (summarized in Table 4):

1. COMPAS (Angwin et al., 2016)[3] attempts to predict recidivism with recidivists comprising 53% of the dataset. For fairness, we consider just two of the provided races (African-American and Caucasian) as sensitive attributes, in which African-Americans are the majority class at 60% of the dataset.

2. Communities and Crime (Redmond, 2009)[4] (Crime) is used for regression to determine if a community will have a high violent crime rate. We adapt this task for classification by binarizing the target *ViolentCrimesPerPop* to $\geq$ .7 and for the fairness attribute binarize *RacepctBlack* $\geq$ .06 in which the disadvantaged communities have higher Black populations and make up 52.1% of the dataset.

3. Census Income (Adu, 1996)[5] (Adult) attempts to predict if an individuals income is $\geq$ $50,000$, in which 75% of the individuals have a salary below 50K. We use gender (Male, Female) as the sensitive attribute, in which Males composed 68% of the dataset.

4. Default Credit (Def, 2016)[6] (Credit) seeks to predict if an individual will default on an credit card payment. The default class composed 22% of the dataset, and the sensitive attribute used is the age with the minority group of young individual under 25 constituting 13% of the dataset.

## G.2  DETAILS ON OUR IMPLEMENTATION

We use two distinct code bases: one for efficient ZK protocol implementation and the second for assessing the accuracy and fairness of our fair decision tree.

**EMP-toolkit.** We use EMP-toolkit (Wang et al., 2016) to implement our ZK protocol. EMP is written in C++ and offers efficient implementations of ZK protocols. This code base is used for timing results (benchmarking the efficiency of our ZK protocol) and conducted using two Amazon EC2 c6a.2xlarge machines to represent the prover and verifier. We use the Linux tc command to simulate a LAN connection between the two machines with a bandwidth of at most 1000Mbit/sec and latency of 2ms. We report the median runtime of 5 experiments at each parameter setting.

**JSAT with Fair Trees and Forests.** The Java Statistical Analysis Tool (JSAT) (Raff, 2017), is an open source Java library which supports decision tree and random forest models. We built on the Fair-Forest extension of JSAT in Fantin (2020) and implemented splitting criteria to recreate the work of Kamiran et al. (2010) and Raff et al. (2018) as a baseline for demographic parity fair trees. Additionally, we implement a splitting criterion for $Gain_{\text{Unfairness}}$ based on thresholds for both demographic parity and equalized odds fairness.

---

[3]Retrieved from https://www.propublica.org/datastore/dataset/compas-recidivism-risk-score-data-and-analysis

[4]Retrieved from https://archive.ics.uci.edu/ml/datasets/communities+and+crime

[5]Retrieved from https://archive.ics.uci.edu/ml/datasets/adult

[6]Retrieved from https://archive.ics.uci.edu/ml/datasets/default+of+credit+card+clients

To evaluate Confidential-PROFITT, we train decision trees and random forests for 250 values of $\tau$ with 10 random seeds each. For each dataset, we set the height of the tree by observing test and training set results in a decision tree trained without fairness. We choose the smallest height that maintains accuracy without overfitting. These heights are reported for each dataset in Table 4.

We evaluate fairness and accuracy using Fairlearn (Bird et al., 2020) and SciPy (Virtanen et al., 2020) over a testing set using a test-train split of $75\% : 25\%$. Experiments are run on 8 Intel Xeon CPUs with Java (v. 14.0.2).

### G.3 DETAILS ON OUR EVALUATION METRICS

**Demographic parity unfairness.** Demographic parity (DP) unfairness is often characterized by the absolute value of the gap between the positive prediction rate for each subgroup. DP unfairness has a range of $[0, 1]$ in which a value of 0 corresponds to perfect DP fairness. We measure this with Fairlearn's fairlearn.metrics.demographic_parity_difference function (Bird et al., 2020).

$$\text{DP unfairness} = \big| \Pr[\hat{Y} = 1 | S = a] - \Pr[\hat{Y} = 1 | S = b] \big| \qquad \forall a, b \in S.$$

**Equalized odds unfairness.** Equalized odds (Eodds) unfairness is characterized by the largest gap between subgroups' true positive and false positive rates. Eodds unfairness has a range of $[0, 1]$, in which a value of 0 corresponds to perfect Eodds fair. We measure this with Fairlearn's fairlearn.metrics.equalized_odds_difference function (Bird et al., 2020).

$$\text{Eodds unfairness} = \big| \Pr[\hat{Y} = 1 | Y = 1, S = a] - \Pr[\hat{Y} = 1 | Y = 1, S = b] \big| \qquad \forall a, b \in S.$$

**Unfairness gain of the trained decision tree.** Unfairness gain ($Gain_{\text{Unfairness}}$) is the information gain over parent and child nodes' Gini impurity ($Gini_{\text{Unfairness}}$) of sensitive attribute $a$. $Gini_{\text{Unfairness}}$ may be taken with respect to demographic parity or equalized odds unfairness.

**Accuracy.** The accuracy is computed from the ratio of the number of correct predictions to the total number of predictions. Here, we use SciPy's sklearn.metrics.accuracy_score function (Virtanen et al., 2020).

## H COMMUNICATION COSTS

In this section, we evaluate the communication costs of our framework.

Figure 8 evaluates the communication costs of Confidential-PROFITT as a function of number of training samples, number of attributes, and height of trees using both demographic parity and equalized odds. The communication cost of Confidential-PROFITT increases linearly as we increase the number of training samples or the number of attributes. The reason for this is that the communication cost of committing to each sample is constant, and this cost increases as the number of attributes increases. The increased steepness of the $10^6$ sample benchmark is likely a result of the batch amortization used by our underlying ZKP framework. In general, equalized odds incurs higher communication costs than demographic parity as proofs need to be performed over four counting trees in equalized odds, but only two in demographic parity.

## I RANDOM FOREST CAN IMPROVE UPON THE PERFORMANCE OF A DECISION TREE

We show results for the model performances of both decision trees and random forests trained without fairness in Table 5. These results show that random forests improves upon the accuracy of decision trees across all datasets. However, random forests are typically more unfair, supporting prior conclusions that standard ensemble strategies alone cannot improve fairness (Feffer et al., 2022; Bhaskaruni et al., 2019).

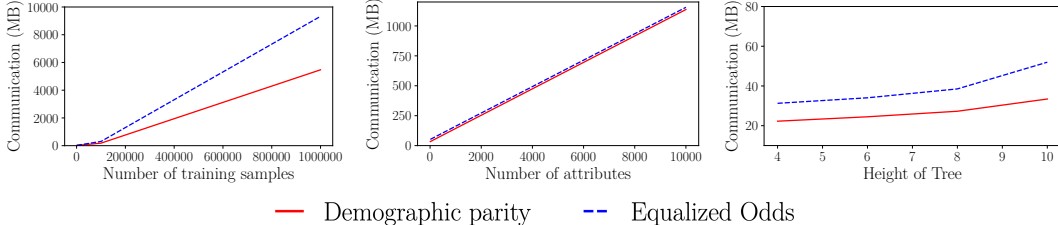

Figure 8: Communication costs of Confidential-PROFITT using demographic parity and equalized odds. Default settings are: $10,000$ for training samples, $10$ for attributes and $10$ for tree height.

| Dataset | Accuracy | | Unfairness (Demographic Parity) | | Unfairness (Eodds) | |
|---------|----------|----|----------------------------------|----|---------------------|----|
| | DT | RF | DT | RF | DT | RF |
| Crime | $.872 \pm .015$ | $.891 \pm .015$ | $.497 \pm .088$ | $.495 \pm .079$ | $.432 \pm .129$ | $.443 \pm .100$ |
| COMPAS | $.601 \pm .023$ | $.652 \pm .022$ | $.121 \pm .057$ | $.178 \pm .097$ | $.135 \pm .055$ | $.190 \pm .097$ |
| Adult | $.759 \pm .016$ | $.864 \pm .002$ | $.102 \pm .018$ | $.130 \pm .010$ | $.076 \pm .014$ | $.091 \pm .019$ |
| Credit | $.980 \pm .003$ | $.985 \pm .002$ | $.042 \pm .011$ | $.034 \pm .010$ | $.028 \pm .014$ | $.026 \pm .019$ |

Table 5: Comparing accuracy and fairness of decision trees versus random forests.

---

**Algorithm 6:** ZK proof of fair training of a random forest.

**Input:** Training set $X$, number of trees in ensemble $N$, number of attributes for each tree $M$, number of training samples for each tree $L$, array of $N \cdot M$ random integers $[r_{1,1}^a, ..., r_{N,M}^a]$, array of $N \cdot L$ random integers $[r_{1,1}^s, ..., r_{N,L}^s]$ (known to both parties).
**Output:** Commitment to random forest parameters, ZK proof that parameters are fair.

1: Prover commits to each row in the training set $[\![X]\!]$
2: **for all** $i \in [1, N]$ **do**
3:     initialize empty sample array $X_i$
4:     restrict attributes in $[\![X]\!]$ to those indexed by $r_{i,1}^a, ..., r_{i,M}^a$.
5:     **for all** $j \in [1, L]$ **do**
6:        add $[\![X]\!][r_{i,j}^s]$ to $X_i$
7:     Prover locally trains a fair decision tree DT with threshold $\tau$ over unfairness-based information gain using $X_i$ as training data
8:     Run Algorithm 2 with inputs $X_i$, DT, and $\tau$
9: Output the concatenated outputs of Algorithm 2.

---

## J  OUR ZERO-KNOWLEDGE PROOF PROTOCOL FOR RANDOM FOREST TRAINING WITH FAIRNESS

Our protocol for the fair training of decision trees is described in Algorithm 6. The parties use the input randomness (decided via a secure coin-flipping protocol) to index into a committed dataset, thus obtaining subsamples. Next, decision trees can be trained on the subsamples to build the trees used in the random forest. Finally, the fairness of the trees can be verified individually using Algorithm 2. Note that we use the same tree heights for random forests as for decision trees (as reported in Table 4). We create random forests from a collection of 10 decision trees in which each node considers a random subset of attributes of size $\sqrt{H}$, in which $H$ is the total number of attributes.

## K  ADDITIONAL RESULTS ON THE RELATIONSHIPS BETWEEN UNFAIRNESS INFORMATION GAIN AND UNFAIRNESS DEFINITIONS

Figures 9 and 10 show the relationship between $Gain_{\text{Unfairness}}$ and the fairness gap for demographic parity and equalized odds, respectively. The $Gain_{\text{Unfairness}}$ reported in these figures was computed from the average values for all nodes of ten trees trained from one of 250 threshold values. They show a positive correlation between $Gain_{\text{Unfairness}}$ and the fairness gap, supporting our intuition that bounding $Gain_{\text{Unfairness}}$ leads to a smaller fairness gap.

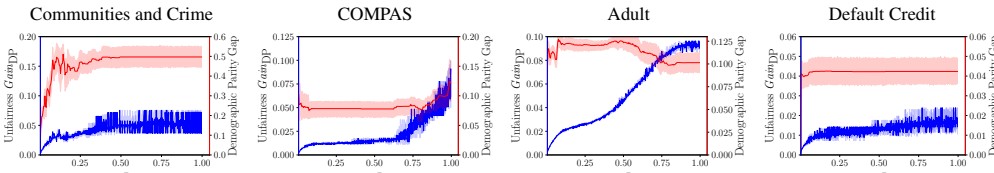

Figure 9: Relationship between unfairness information gain and the demographic parity gap. We compute the average unfairness information gain over all node splits for decision trees created with a sweep of threshold bounds on unfair information gain.

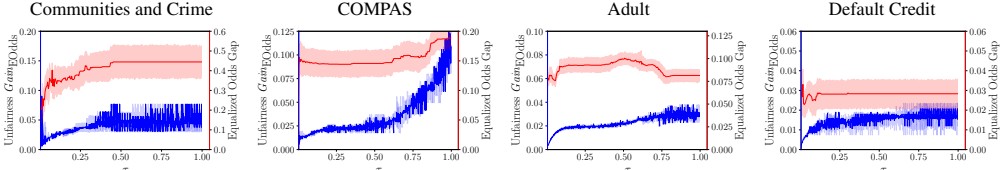

Figure 10: Relationship between unfairness information gain and the equalized odds fairness gap. We compute the average unfairness information gain over all node splits for decision trees created with a sweep of threshold bounds on unfair information gain.

