# OpenReview forum: "Confidential-PROFITT: Confidential PROof of FaIr Training of Trees"
_ICLR.cc/2023/Conference — ICLR 2023 notable top 5%_

### Official Review · Reviewer_hxRN · 2022-10-21

**Confidence:** 3
**Correctness:** 4
**Technical Novelty And Significance:** 3
**Empirical Novelty And Significance:** 4
**Recommendation:** 8

**Clarity, Quality, Novelty And Reproducibility:**

The authors use a state-of-the-art framework for zero-knowledge to present benchmarks.

It would be good if the authors also present figures on the communication or proof size.

I managed to run the code.

Minor comments:
- Erroneous whitespace after footnote 1
- Using sans-serif for the name of the construction decreases readability and makes the paper visually more "noisy".


**Strength And Weaknesses:**

I think this is in a great application of zero-knowledge proofs.


**Summary Of The Paper:**

The paper presents two different ways of training a decision tree with fairness criteria and then providing a zero-knowledge proof certifying the fairness without revealing any further information.


**Summary Of The Review:**

Interesting and well-executed application of zero-knowledge proofs

---

> ### Author Response · Authors · 2022-11-11
> **Authors' reply to reviewer hxRN**
>
> We thank the reviewer for their comments.
>
> > **It would be good if the authors also present figures on the communication or proof size**
>
> Thanks for the suggestion. We evaluated the communication costs of our framework using both demographic parity and equalized odds
>  - for all datasets (see last row in Table 1, Section 4); and
>  - as a function of the number of training samples, the number of attributes, and height of trees (see Figure 8, Appendix H).
>
>
> We described the communication costs of real-world datasets in Section 4 as:
>
> “The communication overhead for both parties is at most 145 MB. For example, the total communication cost of running CONFIDENTIAL-PROFITT using equalized odds and demographic parity on the COMPAS dataset is 28 MB and 21 MB, respectively.”
>
>
>
> |COMPAS (Eodds) | COMPAS (Dem.Parity) | Crime  (Eodds) | Crime (Dem.Parity) | Default Credit  (Eodds) | Default Credit (Dem.Parity) | Adult (Eodds) | Adult (Dem.Parity)|
> |-------|--------|--------|--------|--------|--------|-------|-------|
> |28.7MB | 21.1MB|23.9MB|16.9MB|107.3MB|67.7MB|145.2MB|89.1MB|
>
>
>
>
> We also described the communication benchmarks in Appendix H as follows.
>
> “In this section, we evaluate the communication costs of our framework.
> Figure 8 evaluates the communication costs of CONFIDENTIAL-PROFITT as a function of number of training samples, number of attributes, and height of trees using both demographic parity and equalized odds. The communication cost of CONFIDENTIAL-PROFITT increases linearly as we increase the number of training samples or the number of attributes. The reason for this is that the communication cost of committing to each sample is constant, and this cost increases as the number of attributes increases. The increased steepness of the $10^6$ sample benchmark is likely a result of the batch amortization used by our underlying ZKP framework. In general, equalized odds incurs higher communication costs than demographic parity as proofs need to be performed over four counting trees in equalized odds, but only two in demographic parity.”
>
>
>
> > **Erroneous whitespace after footnote 1**
>
> We fixed the white-space error of footnote 1 as you suggested, thank you for spotting this.
>
>
> > **Using sans-serif for the name of the construction decreases readability and makes the paper visually more "noisy".**
>
> Regarding the sans-serif font, our intention was to improve clarity but given your concern, we have put it back to standard font.

---

### Official Review · Reviewer_iYrA · 2022-10-24

**Confidence:** 2
**Clarity, Quality, Novelty And Reproducibility:** The current paper has good quality, c…
**Correctness:** 4
**Technical Novelty And Significance:** 4
**Empirical Novelty And Significance:** 4
**Recommendation:** 8

**Strength And Weaknesses:**

Strength:
1. The idea of considering confidential proof of fairness for training is interesting.
2. They design and implement the zero knowledge proof protocol for verifying the fairness and efficiency.
3. They conduct experiments for verifying the effectiveness and efficiency of proposed methods.
Weakness:
1. The paper proposed a method for ZK proof of decision tree training process. Also the authors mention that "Our objective in this work is to provide a company with an approach that can be used to prove to a third party the fairness of a machine learning model trained and deployed by that company". It seems that the proposed method does not meet the proposed goal.

**Summary Of The Paper:**

The paper considers achieving the zero knowledge fair decision tree learning algorithms to guarantee confidentiality both the model and training data. An extension to ensembles of trees is also proposed.

**Summary Of The Review:**

The paper proposed a zero knowledge fair decision tree learning algorithm and verify the effectiveness and efficiency via benchmark datasets. Overall, I think the current paper is good.

---

> ### Author Response · Authors · 2022-11-11
> **Authors' reply to Reviewer iYrA**
>
> We thank the reviewer for their thoughtful review.
>
> > **The paper proposed a method for ZK proof of decision tree training process. Also the authors mention that "Our objective in this work is to provide a company with an approach that can be used to prove to a third party the fairness of a machine learning model trained and deployed by that company". It seems that the proposed method does not meet the proposed goal.**
>
> In the last paragraph of page 1, we tried to clarify (in the original version of the paper) that we introduce
>
> “...confidential proofs of fair training. We highlight that our method does not *guarantee* fairness. Rather, our approach employs a tunable parameter controlling the resulting degree of fairness. The resulting certificate proves that our approach was employed, and also includes the specific parameter value used as well as the resulting fairness metrics when computed on the training data. We call this approach *fairness-aware training*, or *fair training* in short.”
>
> We acknowledge that the sentence in Section 2 near the top of page 3 in which we say
>
> “Our objective in this work is to provide a company with an approach that can be used to prove to a third party the fairness of a machine learning model trained and deployed by that company.”
>
> is potentially confusing, and we have revised it, thanks for your careful reading and pointing this out! In our revised version, we instead say:
>
> “Our objective in this work is to provide a certificate proving that a company employs a fair training algorithm with a tunable parameter controlling the resulting degree of fairness.”
>
> We’d be glad to clarify further if you have any follow-up questions.

---

### Official Review · Reviewer_C481 · 2022-10-25

**Confidence:** 2
**Correctness:** 4
**Technical Novelty And Significance:** 3
**Empirical Novelty And Significance:** 3
**Recommendation:** 8

**Clarity, Quality, Novelty And Reproducibility:**

I think this paper is well-written. I am not an expert on this topic so I could not provide a fair judgment on the novelty.

**Strength And Weaknesses:**

Strengths: 1. The proposed method is built upon standard protocols. It is efficient and secure.
2. The proposed method can be easily extended to various fairness metrics.
Weaknesses: 1. The data or the model are not revealed to the auditors. It would be better if the authors could discuss or demonstrate through experiments about the confidentiality from an adversary perspective.

**Summary Of The Paper:**

This paper proposes a fair decision tree algorithm with confidential certification of fairness during training. This paper first provides a novel fair decision tree algorithm and then provides a zero-knowledge proof based verifier using frequency counting. This paper also perform extensive experiments to demonstrate the effectiveness of the fair training and also the auditing of fairness.

**Summary Of The Review:**

This paper provides the first practical and efficient method for auditing fairness during training without revealing data or the model. This paper also provides extensive experiments to verify the accuracy, fairness, running time, and scalability.

---

> ### Author Response · Authors · 2022-11-11
> **Authors' reply to Reviewer C481**
>
> We thank the reviewer for their careful reading and kind comments.
>
> >  **It would be better if the authors could discuss or demonstrate through experiments about the confidentiality from an adversary perspective.**
>
> Our approach relies on zero-knowledge proofs to formally guarantee the confidentiality of the learned model as well as of the data even against a malicious auditor (see Appendix E for the mathematical proof). To be more precise, the confidentiality of the model and data is ensured by the “zero-knowledge” property of the underlying cryptographic protocol that can tolerate any malicious behavior [1,2]. The security of these protocols is in turn based on standard hardness assumptions used in cryptography. We note that the zero-knowledge proof protocol itself is not the focus of this paper; instead, we propose a protocol and training algorithm that leverages existing zero-knowledge proof techniques to efficiently prove fairness-aware training. The confidentiality of our system can be ensured by any secure zero-knowledge proof system (see Appendix E for details), including the one used in this paper.
>
> We highlighted the above discussion in Appendix E of our revised manuscript. Please let us know if we have misunderstood - we’d be glad to respond to any further questions or concerns.
>
> References:
>
> [1] Shafi Goldwasser, Silvio Micali, and Charles Rackoff. The knowledge complexity of interactive proof-systems (extended abstract). In 17th ACM STOC, pp. 291–304, Providence, RI, USA, May 6–8, 1985. ACM Press. doi: 10.1145/22145.22178.
>
> [2] Oded Goldreich, Silvio Micali, and Avi Wigderson. Proofs that yield nothing but their validity or all languages in NP have zero-knowledge proof systems. Journal of the ACM, 38(3):691–729, 1991.

---

### Author Response · Authors · 2022-11-11
**Authors' (general) reply to all reviewers**

Dear all reviewers,

We thank you for noting the strengths of our paper, namely:
- Interesting and original idea;
- Practical and confidential framework for auditing the fair training of a model;
- Extensive experiments verifying the effectiveness and efficiency of our framework;
- Well-written paper;
- Great application of zero-knowledge proofs.
We are also glad that you ran our code successfully.

We respond below to all questions raised and have updated our manuscript, showing changes in Magenta color. We summarize the changes here:
- Added the communication cost of our framework using demographic parity and equalized odds for all datasets in Table 1 of our manuscript, and described in Section 4;
- Analyzed the effect of the number of training samples, number of attributes and height of tree on the communication cost in Figure 8 of our manuscript, and described in Appendix H;
- Discussed and clarified the confidentiality guarantee that our framework provides in Appendix E of our manuscript;
- Revised the sentence in Section 2 of our manuscript near the top of page 3;
- Used standard font for the name of our framework;
- Fixed the white-space error of footnote 1.

If you have any further questions, comments or concerns, we would be glad to respond promptly.

Thank you,

Paper4102 Authors

---

### Decision · Program_Chairs · 2023-01-20

**Decision:**

Accept: notable-top-5%

**Justification For Why Not Higher Score:**

N/A

**Justification For Why Not Lower Score:**

see summary

**Metareview: Summary, Strengths And Weaknesses:**

Very interesting application of zero knowledge proofs, good experiments, well-written, all reviewers strongly in favor of accepting.

**Note From Pc:**

if the above contains the word "oral" or "spotlight" please see: "oral" presentation means -> notable-top-5% and "spotlight" means -> notable-top-25%. As stated in our emails, we are disassociating presentation type from AC recommendations